# Spatiotemporal Variability of Extreme Wave Storms in a Beach Tourism Destination Area

**Daniel Guerra-Medina [1] and Germán Rodríguez [1,2,*]**

1  Departamento de Física, Universidad de Las Palmas de Gran Canaria,
   35017 Las Palmas de Gran Canaria, Spain; dguerramedina@gmail.com
2  Instituto Universitario de Estudios Ambientales y Recursos Naturales, Universidad de Las Palmas de Gran
   Canaria, 35017 Las Palmas de Gran Canaria, Spain
*  Correspondence: german.rodriguez@ulpgc.es

**Abstract:** This study explores the spatiotemporal variability of extreme wave storms around the Canary archipelago, with special focus on the southern coastal flank of Tenerife island, a strategic beach tourism destination of large socioeconomic importance. To this end, experimental and simulated data of winds and waves are used to study the severity, seasonality, and directionality of wave storms with considerable potential to cause significant impact on beaches. Furthermore, tidal experimental records are employed to test the joint occurrence of wave storms and significantly high sea levels. Long-term statistical analysis of extreme wave storms at different locations reveals a complex spatial pattern of wave storminess around the islands and in the southern flank of Tenerife, due to the intricacy of the coastline geometry, the presence of deep channels between islands, the high altitude and complex topography of the islands, and the sheltering effects exerted by each island over the others, depending on the directionality of the incident wave fields. In particular, south of Tenerife, the energy content and directionality of wave storms show substantial spatial variability, while the timing of extreme wave storms throughout the year exhibits a marked seasonal character. A specific extreme storm is examined in detail, as an illustrative case study of severe beach erosion and infrastructure damage.

**Keywords:** extreme wave storms; tidal levels; tourist beaches; beach erosion; infrastructure damage; socioeconomic impacts; Tenerife; Canary Islands

## 1. Introduction

Coastal areas are generally densely populated and have high strategic value from a socioeconomic point of view. However, the coast in general, and beaches in particular, constitute systems with highly nonlinear, complex dynamics and are strongly vulnerable to the individual or joint action of different types of natural hazards, which can lead to erosion or flooding processes with significant negative socioeconomic repercussions (e.g., [1]). This is especially true in the case of areas heavily dependent on beach tourism, such as the Canary Islands and, in particular, the southern flank of the Tenerife island [2]. The islands are a tourist destination of world importance, notably in Europe. The archipelago received more than 15 million tourists in 2019 and, according to the Canary Islands Institute of Statistics [3], tourism visiting the islands generates 32% of the Canary Islands' GDP and 30% of jobs. These percentages are even higher on the southern coast of Tenerife, where a large fraction of tourism received in the islands, some six million tourists per year, is concentrated in a few municipalities.

The great majority of tourists visiting the island are looking to enjoy the climate and the beaches (e.g., [4]). In this sense, it is important to bear in mind that beaches are dynamic complex systems that evolve and change their characteristics depending on the hydrodynamic conditions to which they are subjected, particularly during severe wave conditions. Extreme wave storms represent risky events for the natural environment and

human activities on the coast (e.g., [5]). In particular, they have the potential to produce significant beach erosion episodes in relatively short periods, resulting in loss of sand, beach retreat, and the consequent undesirable reduction in beach dimensions. This kind of impact can be temporal or even semi-permanent, depending on the nature of the beach, but, in any case, extends over variable but considerable periods of time, because post-storm beach recovery by long-period swell onshore sediment transport is generally a slow process. Accordingly, it is evident that the occurrence of severe wave storms may have strong negative impacts on tourism activities.

The effects of wave storms on a beach depend on a substantial number of factors, such as their severity, duration, directionality, and seasonality, as well as on the likelihood of occurrence during periods of high (tidal and non-tidal) sea water level elevation, among others. All of the above highlights the importance of having a good understanding of the space–time variability of the extreme wave storms affecting a given coastal zone, as an essential support tool for the development of management strategies that reduce the socio-economic impacts caused by these abnormally severe events.

The study of wave conditions in the Canary archipelago has focused mainly on the northern and northwestern edges of the islands to explore the potential use of waves as a renewable energy resource (e.g., [6,7]). However, the characterization of long-term wave conditions, including high severity episodes, at the southern flanks has received much less attention, despite its potential negative impact on beaches and surrounding areas and the corresponding socioeconomic implications.

In view of the above, the primary focus of this study is to explore the wave climate on the southern coast of Tenerife, with emphasis on stormy conditions, because of the socioeconomic importance of this stretch of coast. Specifically, we focus our interest on those storms with a remarkable capacity to cause damage, mainly in terms of coastal erosion and/or flooding. As a first step towards achieving this goal, the space–time behavior of the wave climate along the outer edges of the Canary Islands is examined to understand the general wave conditions reaching the coasts of the archipelago.

The paper is structured as follows. After justifying the need to study extreme wave storm characteristics, in both space and time domains, in the southern coastal strip of Tenerife island, a brief description of the geographical and meteoceanic characteristics of the study area, as well as the main characteristics of experimental data used in the study, is provided in Section 2. The methodological approaches used to examine the spatial and temporal variability of extreme wave storms are introduced in Section 3. Section 4 presents the results derived from the study and their discussion, including the detailed study of a selected storm, as an illustrative case of severe beach erosion and infrastructure damage, highlighting the evolution of wind, wave, and tidal conditions, and showing evidence of its socioeconomic impacts. Finally, Section 5 summarizes the main findings of the study.

## 2. Study Area and Datasets

### 2.1. Study Area

The Canary archipelago consists of seven major islands and several islets and constitutes a Spanish autonomous community located on the Northwest African continental shelf, in the Eastern Central Atlantic, off the Saharan coast at a minimum distance close to 100 km, measured from Fuerteventura, the easternmost island. Gran Canaria (GC) and Tenerife (T) are the two most populated islands, together constituting more than 80% of the total population (over 1,750,000 inhabitants). The rest of the residents are mostly concentrated in Fuerteventura (F), Lanzarote (L), and La Palma (P), while the minor islands of El Hierro (H) and La Gomera (G) are barely populated. The archipelago is approximately centered at the coordinates (28° N, 15° E) and extends around 450 km from east to west between 27° and 30° of northern latitude, as shown in Figure 1. The enlarged image shows the relative position of each island within the archipelago and with respect to the African continent, as well as the complex geometry of the islands' coast. In addition, it shows an enlarged illustration of the southern side of Tenerife. An important aspect, not observed in

the figure, is the altitude of the islands. Lanzarote and Fuerteventura, the islands closest to Africa, are significantly flat, but the altitude is over 1500 m in the further west islands, reaching the maximum height (3715 m) at Pico del Teide, Tenerife, the largest (2034 km$^2$) and highest of the islands.

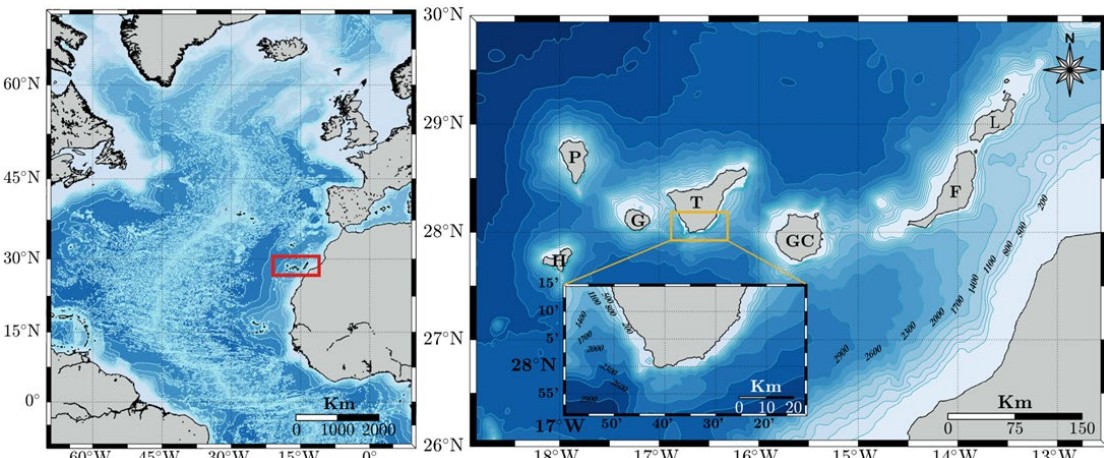

**Figure 1.** Geographical location of the Canary Islands (**left**) and zoom of the archipelago, with the African coast as a reference in the (**right**) lower corner, as well as an expansion of the southern edge of Tenerife island.

Due to its geographical location, in the southern edge of the Azores High, the Canaries are within the fairly regular Trade Winds belt. The trade winds regime exhibits a clear seasonal pattern throughout the year, governed by the relative intensity and location of the Icelandic Low and Azores High pressure systems. During summer, the dominant trade winds blow with moderate or weak intensity from the N-NNE directional sector, with frequencies between 90% and 95%, while, during winter, NE trade winds blow with lower intensity and frequency, over 50% (e.g., [8]).

Naturally, the wave climate in the archipelago is strongly related to the above atmospheric conditions and, consequently, wave conditions are rather mild, Thus, the northern edge of the island is the most exposed to wave action, while there are significant spatial variations around the islands' coastline [9,10]. Furthermore, wave conditions undergo a clear seasonal pattern in the most energetic areas, the northern and western sides of the archipelago, with mild wave conditions from April to October and more severe situations from November to March. Regarding wave storm conditions on the north side of the archipelago, it has been observed that extreme wave events also exhibit a statistically significant seasonal behavior [11]. The tidal regime in the islands is mesotidal, with a semidiurnal tide pattern and a tidal range oscillating approximately between 0.5 m and 3 m, and a mean value close to 1.5 m [12]. Furthermore, meteorological residuals are almost negligible, ranging within ±20 cm, approximately, but with a modal value that is almost null [13].

### 2.2. Datasets

The investigation is based on four datasets of different nature. On the one hand, datasets including oceanographic and meteorological information have been provided by the Spanish Port authority and include wind and wave data obtained from the coupling of wind and wave numerical models (hindcasting), wind and wave observations recorded in-situ by means of meteoceanic buoys, and mean water level information registered with a tide gauge. On the other hand, in the absence of more rigorous sources of information, evidence on the impact of wave storms in the coastal zone of interest has been obtained by using the digital information database JABLE [14], created by the University of Las Palmas de Gran Canaria, which includes an enormous volume of historical and current press produced in the Canary Islands from 1808 to the present. This digital platform allows

searching on a specific topic with keywords by island, locality, period, etc., among more than 7 million pages from more than 700 newspapers, newsletters, bulletins, gazettes, journals, magazines, and other serial publications.

The database containing wind and wave information obtained by using wind and wave numerical models is referred to as SIMAR and provides information covering the period from January 1958 to March 2020. Placement of computational mesh hindcasting grid nodes selected to characterize wind and wave climate is indicated in Figure 2a. The eight points located in the outer edges of the archipelago are hereinafter cited as external points (EPX), where X indicates each specific point. Similarly, five hindcasting grid nodes used to explore wind and wave conditions in the southern flank of Tenerife are designated as internal points, denoted by IPX. The location of IPX points is more clearly depicted in Figure 2b, which also shows the position of two meteoceanic buoys measuring wind and wave conditions, one located northwest of Gran Canaria and other south of Tenerife, respectively referred to as BGN and BTS wave buoys, and belonging to the network of meteoceanic deep water buoys (REDEXT) of the Spanish Port Authority. These buoys are anchored in areas deeper than 200 m and are located at positions virtually coincident with two SIMAR points. Time series provided by both buoys have an hourly sampling rate and cover the periods from June 1997 to December 2019 (BGN) and from April 1998 to March 2020 (BTS), but directional sensors were not available until 2003. Regarding mean sea level fluctuations, measurements have been carried out with a tide gauge at the northeastern tip of Gran Canaria (TG), as shown in Figure 2b. The corresponding dataset includes hourly values of sea water level and cover the period from July 1992 to March 2020.

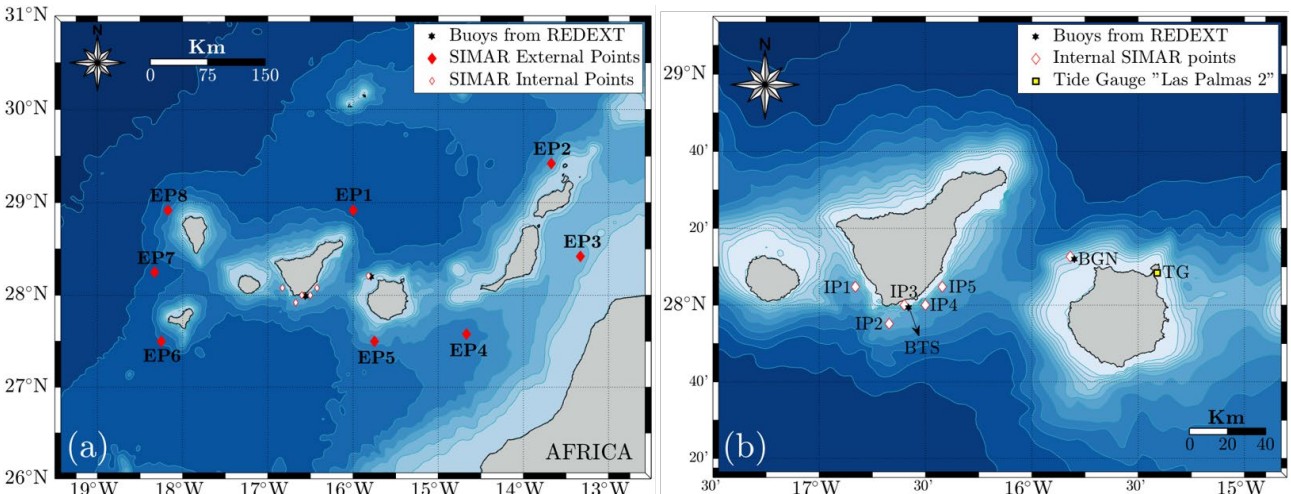

**Figure 2.** Map of the Canary archipelago showing location of external (**a**) and internal SIMAR points, as well as wave buoys and tide gauge (**b**).

Regarding the geometric configuration of the archipelago, it is important to visualize the existence of two deep channels between La Gomera and Tenerife, and between Gran Canaria and Tenerife (Figure 2b), hereinafter referred to as G-T and GC-T channels. The approximate average depths of both channels are 1600 m (G-T) and 2700 m (GC-T), while the minimum widths are 29 km (G-T) and 61 km (GC-T), approximately. It is also important to mention that these three central islands are substantially high, with altitudes exceeding 1500 m and reaching up to 3700 m, approximately.

Standard sea state parameters used to characterize wave climatology have been derived from spectral moments of the directional spectral density function, $S(f,\theta)$, which represents the energy contribution of any wave component to the measured wave field in terms of the frequency, $f$, and propagation direction, $\theta$. The most important parameter for the characterization of a sea state is the significant wave height, $H_{m0}$, defined as four times the square root of the zero-th order spectral moment, $m_0$, which represents the total energy of the process. Therefore, $H_{m0}$ is proportional to the energy content of the corresponding

sea state and, consequently, it is the parameter used to represent its severity, as a general rule. In the case of wave periods, there are several optional characteristic periods that can be used in light of the objectives pursued. Two of the most widely employed in practice are used in this study. These are the average wave period, $T_{02}$, and the spectral peak period, $T_p$, which is the period associated with the most energetic spectral wave components. In this regard, it is important to note that, in contrast to $T_{02}$, $T_p$ is not computed by means of spectral moments but considering a single spectral estimate and therefore presents considerable statistical uncertainty, or statistical variability [15]. In terms of wave direction propagation, the most common parameter used to characterize the directional properties of a wave field is the mean spectral direction, $\theta_m$, which represents the mean approaching direction averaged over all the frequency bands in the directional wave spectrum. The analytical expression of the above-described characteristic parameters in terms of the directional spectrum can be found throughout the literature (e.g., [16]).

## 3. Methodology

The study has been performed at two different spatial levels. On one hand, wave climate in the outer flanks of the archipelago has been explored to identify the characteristics of wave fields reaching the zone, the second one regarding specifically the southern coastal zones of Tenerife island to examine wave climate and the characteristics of extreme wave storms in this coastal stretch in greater detail.

### 3.1. Wave Storm Concept and Definition

Wave storm is a concept that is intuitively easy to understand but difficult to formally define. To a large extent, this is because a storm is a relative term referring to a period of time during which the severity of wave conditions is significantly intense with respect to the conditions normally observed at that location. In this context, wave storms represent extreme events with low frequency of occurrence but potentially severe impacts.

In terms of the above, there is no universally accepted procedure for the identification of wave storms. However, from a practical perspective, a wave storm is commonly considered a sequence of sea states with significant wave height exceeding a given threshold ($H_t$) for a specified minimum time period selected, known as the minimum storm duration ($D_{min}$), so that when the duration ($D$) of an exceedance of $H_{m0}$ is smaller than $D_{min}$, the event is not considered a storm. In addition, it is assumed that two consecutive events must be considered a single storm if the significant wave height in the time between these events does not drop below $H_t$ during a period larger than a certain minimum time interval, usually named the minimum inter-storm duration ($ISD_{min}$), where the time interval between the end of one storm and the beginning of the next is called the inter-storm period, or duration ($ISD$) (e.g., [17]). These parameters are schematically illustrated in Figure 3.

The above definition is not completely rigorous and free of drawbacks. The main problem is the selection of a threshold level that satisfies the criterion of independence of events and at the same time allows the identification of a sufficiently large number of events for the sample to be statistically representative. For this reason, there is great variability in criteria to establish this parameter, depending mainly on the geographic location and the local average wave climate [18].

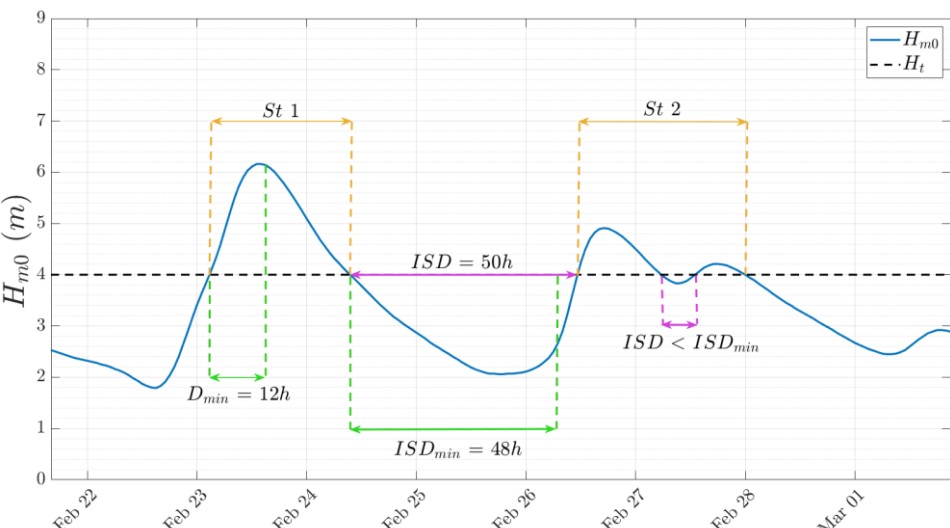

**Figure 3.** Illustrative sketch of parameters used to define and select independent wave storms.

### 3.2. Data Validation

The need for data providing information on long-term wave conditions over extensive areas makes it necessary to use alternative sources of information to that provided by measuring instruments, which are generally scarce, if any, and cover short periods of time. In this sense, the joint use of wind and wave numerical prediction models to obtain long-term databases on a spatial grid is currently a very common and useful tool. However, due to various reasons, wind and wave prediction models have some limitations in their ability to reproduce real conditions, mainly in areas with complex topography, such as islands (e.g., [19,20]). Consequently, whenever possible, it is necessary to validate the results against other sources of information, preferably from specific experimental measurement devices, such as meteoceanic buoys. Data validation has been achieved in this study by comparing data derived from models at two nodes, located at positions coinciding with that of BGN and BTS wave buoys (see Figure 2b), and data measured by these buoys, considering only the periods for which measured data were available.

Due to the interest in extreme wave storms and considering the difficulty of models to correctly reproduce extreme wave conditions [21,22], characteristic parameters associated with each selected wave storm were validated by linear regression and graphically represented for visually assessing the degree of agreement between simulations and experimental recordings. Moreover, joint validation of circular (wind and wave direction) and linear variables (wind speed, wave height and period) has been done by examining wind and wave roses for measured and simulated data.

### 3.3. Identification of Storms with Severe Impacts on Beaches

Digitized newspaper archives covering large time periods constitute a rich source of socioeconomic and environmental information. In particular, archives of regional newspapers are of great value for identifying the occurrence of non-recorded past and present extreme events of natural phenomena, as well as for obtaining a rough idea of their damage intensity and their socioeconomic impacts, which is of great importance for coastal managers [23,24]. Nevertheless, despite its considerable usefulness in this sense, this type of information should be considered with due caution [25]. With this in mind, a Boolean search with different combinations of keywords was developed in the JABLE database to detect dates on which the press reported wave storms causing damages on beaches, or nearby infrastructures in the study area, to be used as a source of evidence of the impact of severe damaging events.

### 3.4. Wave Storm Severity

Several parameters have been suggested in the literature to assess the severity of wave storms. The Storm Power Index [26] was introduced as the product of the squared maximum value of $H_{m0}$ during the storm and its duration. Clearly, this parameter overestimates the storm severity by considering only the maximum of $H_{m0}$ during the storm. Accordingly, an integral parameter to quantify the total wave power, *TWP*, has been proposed [27] and can be expressed as

$$WP = \int_{t_i}^{t_f} H_{m0}^2(t)dt \approx \Delta t \sum_{t_1}^{t_2} H_{m0}^2 \tag{1}$$

where $t_i$ and $t_f$ are the starting and ending times of the storm event, so that the storm duration, *D*, is given by $D = (t_f - t_i)$.

It is interesting to remark that *TWP* is a function of the storm duration and therefore does not allow comparison of the severity of storms with different lengths. To this end, it is appealing to standardize this value with respect to *D* to obtain the storm energy [28], denoted as *E* and given by

$$E = \frac{1}{D} \int_{t_i}^{t_f} H_{m0}^2(t)dt \approx \frac{\Delta t}{D} \sum_{t_1}^{t_2} H_{m0}^2 \tag{2}$$

### 3.5. Wave Storm Seasonality

To assess the existence of seasonal variations in the timing of wave storms throughout the annual period, the day of the calendar year on which the maximum significant wave height of a given storm occurred has been converted to an angular value, $\theta$, assuming that the number of days per year is 365. Thus, in leap years, the data corresponding to 29 February have been removed [11].

To examine whether storms in a given region exhibit a seasonal pattern, it is necessary to know whether it is statistically possible to accept that the time of occurrence of wave storms throughout the year is uniformly distributed. The acceptance or rejection of the uniformity hypothesis is assessed in this study through the use of the Rayleigh and Kuiper tests, by considering the storm peaks' timing throughout the year as a circular variable. The selection of these two tests from among the multitude of existing alternatives is due to the fact that the Rayleigh test is powerful only when it is possible to assume that the population distribution has only one mode, while the Kuiper test is specifically indicated in the case of multimodal distributions. More detailed information on these and other uniformity tests can be found in [11,29] and references therein.

### 3.6. Wave Storm Directionality

Directional characteristics of wave conditions observed or simulated, both at outer and inner points of the archipelago, have been explored by elaborating wind and wave roses. This type of representation in polar coordinates allows an easy visualization of the directional distributions for characteristic wave heights and periods. In particular, the bivariate empirical distributions of the following pairs of characteristic wave parameters have been obtained, $\theta_m - H_{m0}$, $\theta_m - T_{02}$, and $\theta_m - T_p$, for both the total dataset and the values associated with selected storms, at each of the selected points.

## 4. Results and Discussion

### 4.1. Data Validation

Regression analysis during wave storm conditions reveals that values of the regression coefficient, $r^2$, for $H_{m0}$ are quite good, both in the north and south points, although with slightly higher values in the north (0.66 for BGN) than in the south (0.62 for BTS). Figure 4 shows three examples of the significant height evolution, both measured and modeled, during storm conditions. It can be observed that, even with significantly high values of $r^2$, models may overestimate (panel a, $r^2 = 0.82$) or underestimate (panel c,

$r^2$ = 0.87) the experimental measures, although, often, the degree of correspondence is quite good or very good in some cases (panel b, $r^2$ = 0.92). In this sense, it is interesting to underline that although the general trend of the models is to underestimate experimental observations during extreme events (e.g., [16,17]), the results in this study include cases in which storms are underestimated, overestimated, as well as events considerably well reproduced, especially on the northern side of the islands. These results should not be surprising taking into account factors such as the altitude and complexity of the archipelago's topography, the irregularity of the coastline, and the islands' self and mutual shading effects, among others, which limit the ability of models to correctly reproduce wind and wave conditions in these areas.

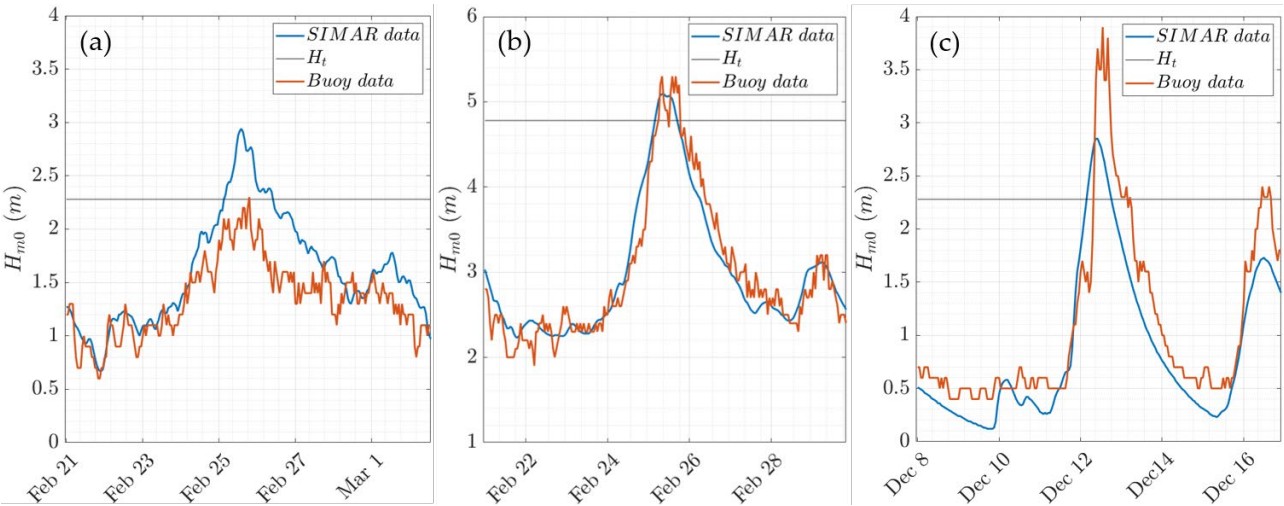

**Figure 4.** Examples of simulated and measured $H_{m0}$ sequences during wave storm episodes measured and simulated at BTS (**a**,**c**) and BGN (**b**), with corresponding regression coefficients $r^2$ = 0.82 (**a**), $r^2$ = 0.92 (**b**), and $r^2$ = 0.87 (**c**).

Regarding the simultaneous evolution of wind and wave directions, the analysis of both parameters as obtained from the buoy and by the model, during storms shown in Figure 4a,b, reveals that there is a fairly good correlation between the experimental measurements and the simulations. A quantitative measure of the correlation between two circular variables can be obtained by means of the circular correlation coefficient, $\rho_c$ [29]. The value of this coefficient for wind measured and simulated directions observed at BTS during the storm of Figure 4a is 0.29, while for the storm detected at BGN during the storm depicted in Figure 4b is 0.78. On the other side, the circular correlation coefficients for wave measured and simulated directions in these two cases are 0.59 and 0.88. In other words, there is a better degree of correlation between wave direction measurements and simulations than between wind direction measurements and simulations. Moreover, the correlation coefficient between both circular variables is higher in the north than in the south. These results can be explained by the lower directional variability of waves than that of wind, as well as the superior ability of the models to simulate wind and wave conditions to the north than to the south of the islands, due to the complexity of their orography. Unfortunately, there is no directional information for the latter case (Figure 4c) since it occurred prior to 2003, the date on which the directional sensors were incorporated into the buoys.

An overview of wind and wave directional variability, as well as their respective combined variability with wind speed, significant wave height, and peak period, can be qualitatively explored by means of wind and wave roses, as shown in Figure 5, for both measured and simulated data at point IP3, south of Tenerife. Panels on the left and in the middle show overall good agreement between average measured and simulated wave direction, although the models slightly overestimate directional dispersion around the mean. Regarding wave height (panels a and d) and period (panels b and e), it can be

observed that, in general, the wave model tends to slightly overestimate the significant wave height while weakly underestimating the peak period.

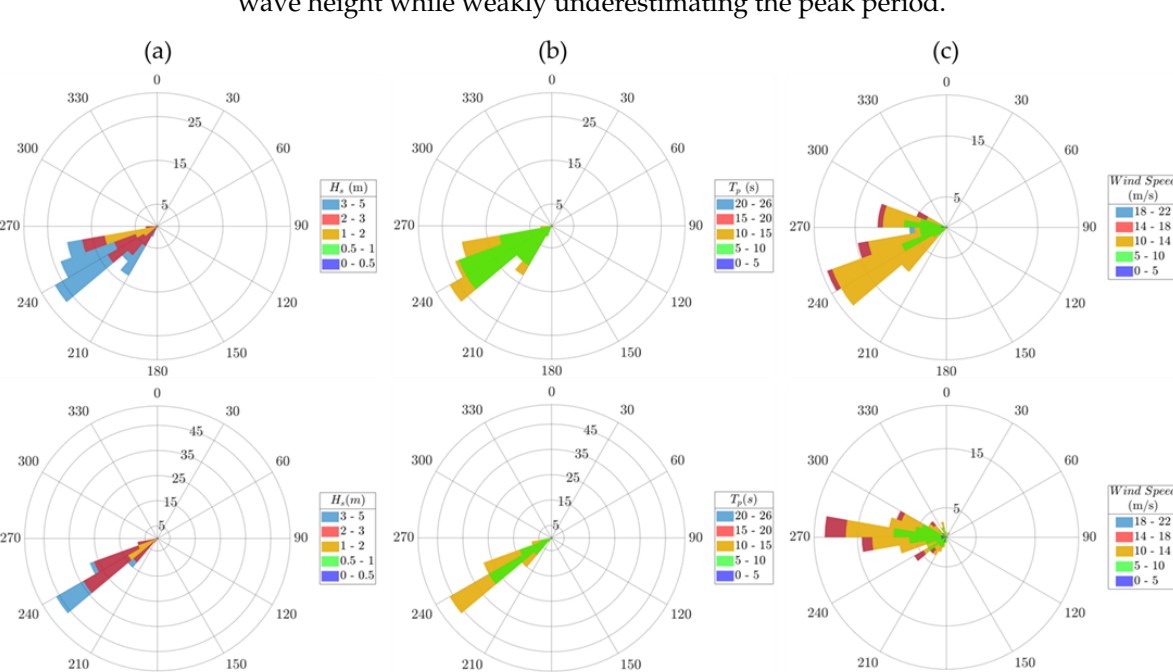

**Figure 5.** Directional distributions of $H_{m0}$ (**a**,**d**), $T_p$ (**b**,**e**) and wind speed (**c**,**f**) during wave storm periods for simulated (IP3, upper plots: (**a**–**c**)) and measured (BTS, lower plots (**d**–**f**) data.

The aforementioned increment in directional dispersion is more noticeable in the case of wind (panels c and f), revealing some weaknesses of the atmospheric model for reproducing wind conditions south of the archipelago during stormy conditions. However, despite the above differences, mostly due to large orographic complexity, the average wind direction and wind intensity are reasonably well reproduced. It is important to note that these effects are much less important in the north of the islands, and are substantially attenuated, both north and south, when all data, not only those of stormy scenarios, are considered. In conclusion, even though the models exhibit some weaknesses, there is overall fairly good agreement between wave measurements and simulations, so that the model results may be used to explore the spatiotemporal variability of wave storms in the study area, although not without some caution.

### 4.2. Wave Storm Parameter Selection

As discussed above, the selection of the optimal threshold is an open issue. Consequently, there are several procedures for selecting an appropriate threshold value. Among these, one of the most widely used in practice is the percentile method, which, like the others, incorporates a certain degree of subjectivity but has the advantage of simplicity. Furthermore, some authors attribute a fairly good degree of robustness to this procedure, depending on the specific use (e.g., [30,31]).

In line with previous comments, the approach used to select the threshold, $H_t$, has been conditioned by three main factors. The first is the specific interest of the study in storms capable of causing substantial damage on sandy beaches. The second is the knowledge of the prevailing wind and wave conditions in the archipelago, and the third is the observed general slight overestimation of significant wave height by the model, mainly during stormy conditions. Accordingly, after trying different quantiles ($Q_{95.0}$, $Q_{99.0}$, $Q_{99.5}$ and $Q_{99.9}$) to select a threshold considering these aspects, it has been observed that using percentiles lower than or equal to $Q_{99.5}$ led to the identification of a substantially high number of storms for an area with a rather moderate wave climate and subjected to

strong sheltering effects against the most frequent storms, generally coming from the NW directional sector, as will be discussed in Section 4.5. Finally, the quantile $Q_{99.9}$ has been identified as the most appropriate threshold for the objectives of the study.

Considering the importance of the occurrence of wave storms coinciding with high tide conditions, the minimum allowable temporal distance between storms, $ISD_{min}$, was set at 48 h, which includes four low and high tide levels and is close to the average duration of atmospheric disturbances in the North Atlantic [11,17]. In the same vein, the minimum duration of the storm has been stated as the duration of a complete tidal cycle, $D_{min}$ = 12 h.

### 4.3. Wave Storm Severity

The severity of wave storms has been evaluated by estimating the total wave power, *TWP*, as in Equation (1), and storm energy, *E*, as in Equation (2). Results obtained at the outer flanks of the Canary Islands for the period covered by the datasets are shown in Figure 6a. Numbers next to each point indicate the number of identified storms, while values in brackets stand for the threshold used to select the events, corresponding in each case to the associated $Q_{99.9}$. It can be observed that the highest values of *E* and *TWP* are located on the western (EP6, EP7, EP8) and northern (EP8, EP1, EP2) flanks of the archipelago, since these are the sides most exposed to harsh wave fields reaching the islands from directional sectors with north and/or west components. On the contrary, the average energy and total wave power of wave storms detected south or east of the archipelago are substantially lower due to sheltering effects against the direct action of storms approaching from any directional sector, except for those travelling from the SW and S sectors. Points EP4 and EP3 are special cases. The first is located at the south but in the middle of the channel formed by Gran Canaria and Fuerteventura, through which waves coming from the NNW-NNE can propagate. Point EP3 is placed at the eastern side of the archipelago, so that wave directions at this point are restricted to NE-SW. It should be noted that proximity to Africa imposes significant fetch restrictions for wave propagation from the east.

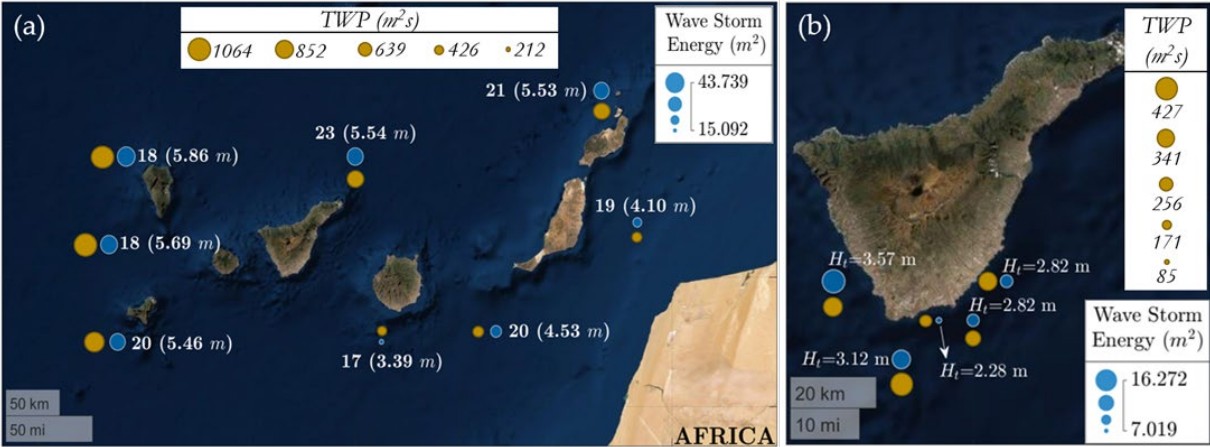

**Figure 6.** (**a**) Average wave storm energy, *E* and *TWP*, at the outer flanks of Canary Islands (1958–2020). Numbers next to each point indicate the number of identified storms and, in brackets, the threshold used to define such storms. (**b**) Average values of *E* and *TWP* at inner points south of Tenerife for the same period. Numbers next to each point indicate the threshold used to define wave storms.

It is worth mentioning that the thresholds established at each point to select the storms follow the same pattern as the average value of the storm energy. However, this is not the case for the number of wave storms, which is very similar for all the outer locations around the archipelago. Nevertheless, this result is totally consistent since wave storms are defined as a function of the threshold that changes from point to point.

Storminess conditions for inner points located at the southern coasts of Tenerife are shown in Figure 6b, which indicates both the storm energy and total wave power, as well as

the selected storm threshold at each point. Results reveal that the average storm energy and total wave power are larger in the southwestern and southern strips, exposed to relatively severe wave storms approaching from the southwest sector, as well as to storms travelling through the G-TF channel from the northwest. It must be stressed that, on this edge of the island, in the vicinity of points IP1 and IP2, are located the most famous beaches of the island. Wave conditions change substantially in the southeastern strips because these areas are protected against wave storms approaching the islands with northern, western, and even southern components. Energy reaching these points comes almost exclusively from the NNE-NE direction, coinciding with the predominant direction of the relatively weak trade winds.

### 4.4. Wave Storm Seasonality

The polar plot shown in Figure 7 presents the date of occurrence for each storm at locations south of Tenerife, with the bubble size indicating storm energy, *E*. It is easy to observe the existence of two distinct climatic seasons, one remarkably mild, from April to October, and other relatively stormy, which extends from November to April, although the period of occurrence for more energetic storms is restricted to a shorter time span, from December to March, approximately. This is the period when severe storms are detected in the southernmost places, while extreme storms reach the southwestern locations from November to April, and the southeastern coastal strip is influenced by less severe wave storms from mid-November to mid-April. An interesting, but rare, feature is the detection of two relatively moderate energy storms at location IP5 during summer. These storms are associated with intense trade wind events, generating moderately severe wave conditions and arriving at this location traveling from the NNE-NE, through the GC-T channel.

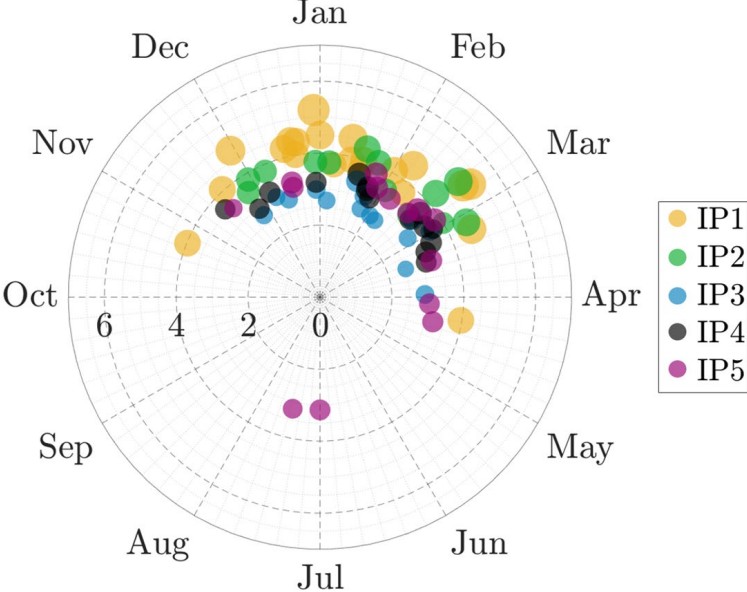

**Figure 7.** Polar plot of $(H_{m0})_{max}$ and the timing of its occurrence throughout the year at all inner points in the southern flank of Tenerife, for the period 1958–2020. Bubble size indicates storm energy.

Generally speaking, extreme storms occur mainly during winter, between December and March, with a substantially lower probability of presentation during spring and autumn, becoming practically null in summer. Accordingly, Rayleigh and Kuiper tests clearly reject the hypothesis of uniformity at any level of significance. In other words, extreme storms are not randomly distributed around the circle (i.e., along the year) but concentrate during late autumn, winter, and early spring, with the more intense conditions observed during winter. Henceforth, the used tests allow us to accept the existence of a seasonal pattern in the timing of extreme wave storms on a statistical basis. It is interesting to note that the tests indicated above have been applied to be rigorous in the analysis,

although, in this particular case, the simple observation of Figure 7 provides qualitative confirmation of this fact.

*4.5. Wave Storm Directionality*

Wind and wave directionality in the external locations is shown in Figure 8. The left panels depict the directionality of wind and wave conditions considering the whole datasets, while the right panels illustrate directionality during selected storm events. The upper panels reveal that the predominant wind conditions in the outer flanks of the archipelago are associated with the prevalence of trade winds, giving rise to relatively mild wind conditions blowing principally from the NNE sector (panel a). However, during severe wave storms, the prevailing wind conditions change significantly, except for the eastern strip (EP3), where the average direction remains from NNE, with a very small directional spread, but the intensity increases notably (panel b). Wind at locations in the northern and western flanks become dominated by intense winds blowing from the WNW-NNW sector. It is interesting to note that winds at locations north of Lanzarote (EP2) display mixed conditions, receiving the influence of the trade winds, intensified during the summer, and the arrival of strong winds from the NW during winter. At the southern location (EP5), wind conditions during wave storm situations become dominated by W-SW intense wind conditions, while point EP4, located in the channel between GC and F, receives the influence of winds with western, northern, and northeastern components.

With respect to the directionality of wave fields reaching the islands at the outer edges and their severity (middle panels), average directional conditions depict a similar pattern to that of the wind, with the predominance of low and moderate wave conditions travelling mainly from the N-NE directional sector, but rolling towards NW when shifting away from the African coast along the northern side and especially on the western coast (panel c). This effect becomes very clear during stormy wave conditions (panel d). In these situations, severe wave storms travel from the NW sector, affecting predominantly the western and northern sides of the archipelago. Due to self-sheltering effects, locations at the eastern side receive much less energy, with waves travelling from the NNE, while southern locations are affected by mild or moderate storms, following a similar pattern to that of the winds, including its propagation through the channels formed between islands.

Regarding the bivariate distribution of wave direction and period (lower panels), the situation is almost similar to that of wave height and directionality. Thus, during stormy conditions (panel f), waves approaching northern and western locations have notably long periods, revealing a swell structure. However, locations sheltered from these conditions (EP3 and EP5) receive smaller and shorter waves, indicating the frequent presence of wind-driven seas travelling from the NNE (EP3) and from WSW (EP5).

In brief, it is worth noting that most frequent and severe storms arrive from the N-NW sector and affect mainly the western and northern sides of the archipelago. The eastern flank is subjected to a strong sheltering effect against these events, so that it is almost exclusively exposed to wave fields generated by the trade winds (N-NNE). The presence of channels between islands allows the propagation of NW-NE wave fields towards the eastern and western edges of the central islands' southern flanks. In the southern areas, during storm conditions, waves approaching from the S-SW sector predominate, although these are usually more moderate than on the rest of the flanks.

Even taking into account the above-mentioned uncertainty associated with the reproducibility of some parameters during extreme events with models, the differences are substantially large and consistent, showing meaningful and realistic changes between the characteristics of the wave fields under general conditions and during extreme episodes.

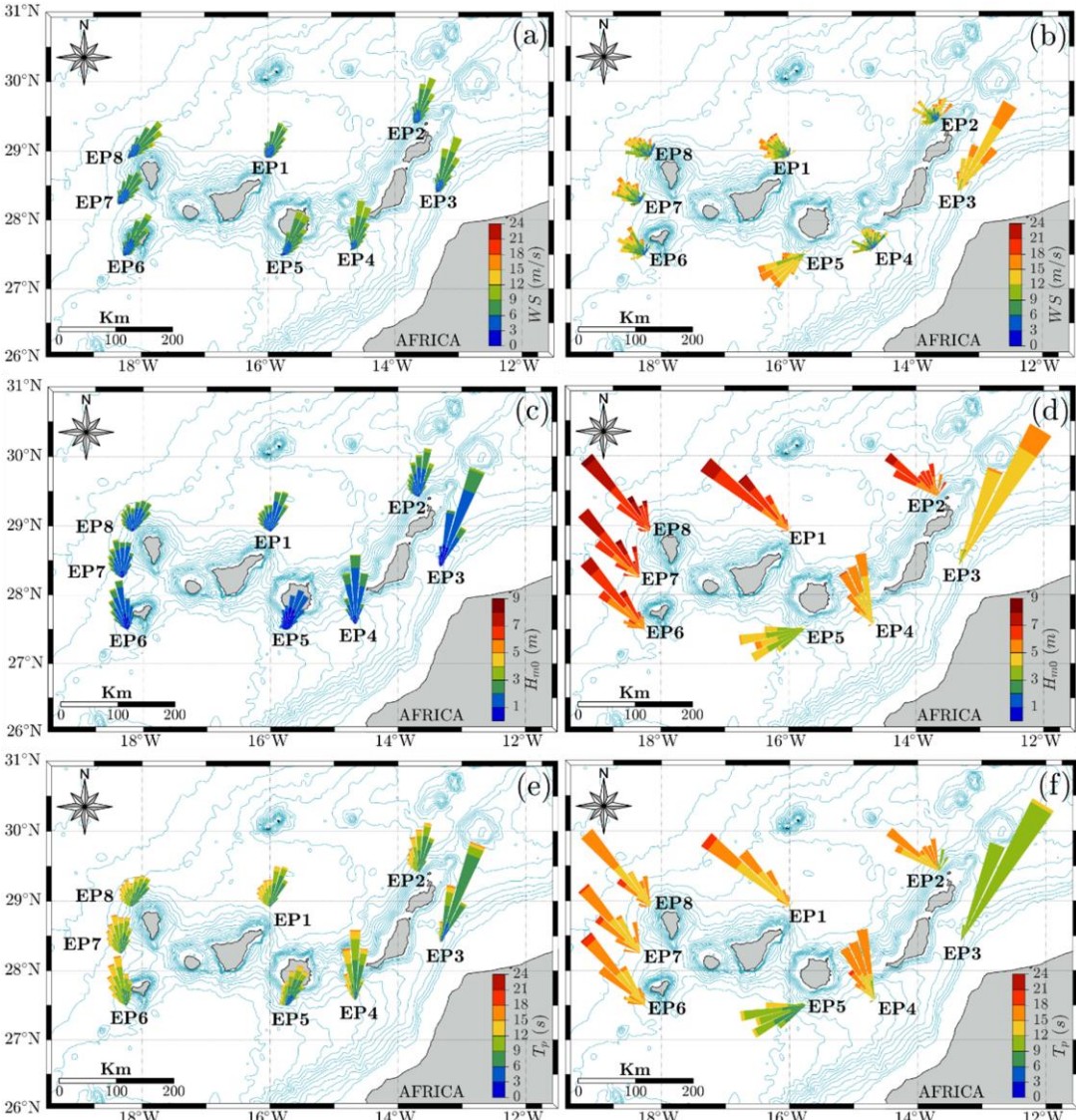

**Figure 8.** Wind and wave roses at outer locations. Wind roses for the whole dataset (**a**), during stormy conditions (**b**). Wave roses for $H_{m0}$ during all the studied period (**c**) and under wave storm conditions (**d**). Wave roses for peak wave period during the whole study interval (**e**) and during storms (**f**).

Results from the analysis of wind and wave directionality at the inner points, located along the southwest, south, and southeast areas of Tenerife, in terms of wind speed, $H_{m0}$, and $T_p$, are shown in Figure 9. Variations in each pair of values are examined under two different types of situation: on the one hand, under average conditions, or average regime, extracted by using the whole dataset, and, on the other hand, during the extreme events selected in each location. Figure 9a,d show that, in general, the dominant direction of the wind regime is NE. Wind speed reaches relatively low values very often (panel a). In contrast, wind speeds observed during wave storms are substantially higher and directionali-ty exhibit a more complex spatiotemporal pattern. In the southwest locations of the island (IP1), the dominant wind direction during wave storms (panel d) shows a considerable dispersion in the NW-NNW sector, although the intensity of the wind is notable only during periods when the wind flows from the NNW, passing through the G-T channel. At points located on the southern flank, the intensity is notably higher than the global average, but the direction in which the wind flows is the opposite, highlighting the presence of strong wind events from the SW. On the other hand, in the southeast

locations (IP4, IP5), the predominant direction remains around the NW, but with strong winds blowing along the GC-T channel.

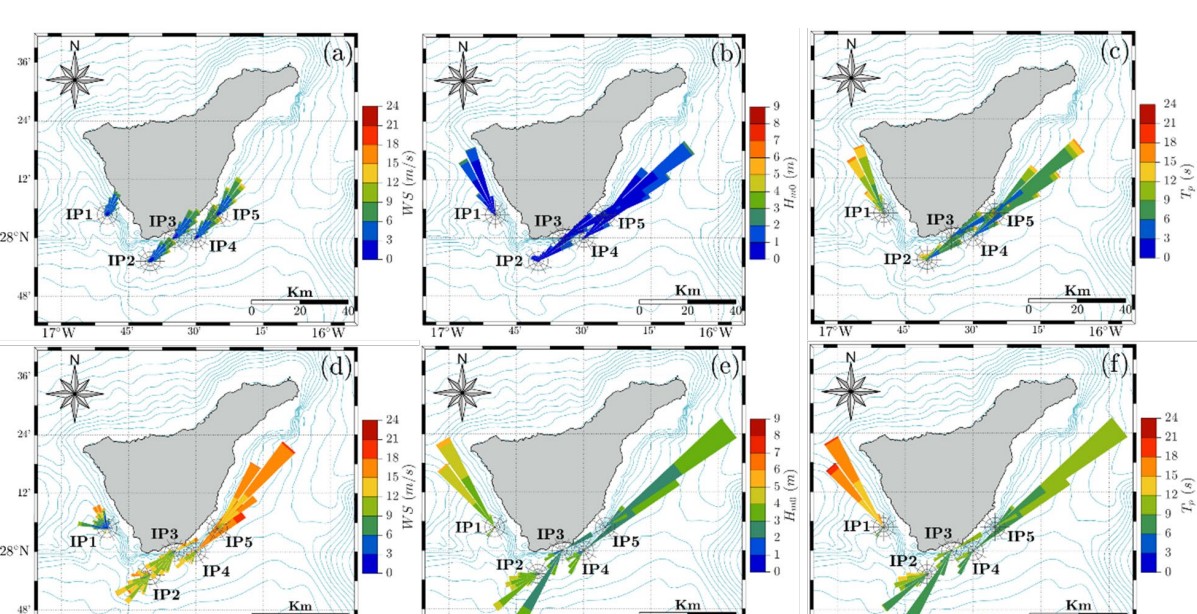

**Figure 9.** Wind and wave roses at inner locations. Wind roses for the whole dataset (**a**), during stormy conditions (**d**). Wave roses for $H_{m0}$ during all the studied period (**b**) and under wave storm conditions (**e**). Wave roses for peak period during the whole study interval (**c**) and during storms (**f**).

With regard to wave conditions south of Tenerife, it should be noted that, under global conditions (panel b), the southern coasts rarely experience wave conditions with $H_{m0}$ values above 2 m. The most western point (IP1) receives waves from the NNW, while waves reaching the southern (IP2, IP3, IP4) and eastern points (IP5) travel predominantly from the NE. Under severe conditions (panel e), the general pattern of wave directionality is practically the same as for wind in these conditions (panel d). However, note that at the westernmost point (IP1), the directional dispersion observed for the wind disappears, with waves travelling almost exclusively from the NNW, through the G-T channel.

The joint distribution of wave direction and peak period reflects an average behavior (panel c) very similar to that observed for wave direction and height. However, it is interesting to note that $T_p$ values associated with storms observed at south and southeast areas (panel f) correspond mainly to locally wind-driven seas or young swell waves. In particular, waves reaching point IP5 and IP4 are associated with dominant trade winds and are funneled through the GC-T channel. Wave storms detected in the southwest points (IP1) are often due to wave fields reaching these coasts from distant storms located in the northwest area of the North Atlantic, through the channels between G-T, while those located in the southern border (IP2, IP3) show a much more complex pattern generated by the alternation of storms arriving through G-T and GC-T channels, as well as less frequently from the SSW-S directional sector.

### 4.6. Illustrative Wave Storm Event with Severe Impacts

With the aim of characterizing in more detail wind and wave conditions during storm events on the southern edge of Tenerife, the variability of wind and wave conditions during the selected extreme events has been examined considering the evolution of wind speed and direction, as well as wave height, period, and direction. Furthermore, mean water wave elevation data measured at the tidal gauge north of GC (TG) have been used to explore the contribution of this phenomenon to the impacts of selected extreme wave storms. Although the analysis was carried out for each of the storms identified south of Tenerife, the evolution

of the parameters during one selected storm is briefly described below as an example. The selection of this storm has been made on the basis of several storm characteristics (storm energy, maximum value of $H_{m0}$ during the event, storm duration, wave direction during the storm) and the existence of evidence of its impact on tourist beaches extracted from the local press historical archives.

The selected storm occurred during late February and early March 2018, and it has been selected principally because of its significantly long duration (around 90 h). The maximum significant wave height during the storm was 4.1 m and the associated return period 12.3 years. This storm, named Emma, began to develop around 23 and 24 February to the NW of the Canary Islands, and progressively increased in intensity to reach its maximum between 26 February and 01 March, while it remained more or less stationary over the archipelago. Then, it continued its trajectory towards Northern Europe, progressively moving away from the islands. This low-pressure system resulted in strong winds blowing from the W-SW sector towards the islands and generating a wave storm that affected almost all the islands and, in particular, the southern and southwestern areas of Tenerife, with a large impact on the tourist beaches located on this coastal stretch.

Figure 10 shows the wave storm evolution as observed at point IP2. Figure 10c shows that before 23 February, the wind direction presents a remarkable variability, until 23 February, when it was established in the S-SW sector, as with the wave direction. On the other hand, Figure 10a reveals a rather positive relationship between the temporal evolutions of wind speed and wave height, indicating that it was a locally generated wave storm. This is also evidenced in Figure 10b, in which it can be seen that both the average and peak periods adopt quite low values during the episode. Finally, in Figure 10d, it can be seen that the storm's peak coincided with a fairly large high tide, with a tidal range of around 2 m. Regarding the effect of tidal level on the wave storm impacts, the evaluation of tidal ranges coincident with the timing of storm occurrence, during the period when tidal information is available (1992–2020), reveals that the tidal range during the storms identified as damaging events was, in all cases, higher than the average value (1.5 m), pointing out the well-known significant contribution of this factor to the impact of wave storms on the coastline. In this particular example, the combination of both phenomena caused important negative impacts, including damage to coastal structures and buildings near the coastline, as well as erosion problems on the beaches of Arona (Southwestern Tenerife), which were reported by the local press, such as observed in Figure 11, which shows the lack of sand and the severe damage caused on the promenade of Los Tarajeles beach, in Los Cristianos (Arona), which had to be closed during the period of their repair. Furthermore, the "Francisco Andrade Fumero" promenade, on Las Américas beach, was closed to public use due to flooding and damage. During the stormy period, the red flag was present on all the beaches of the highly tourism-dependent municipality of Arona.

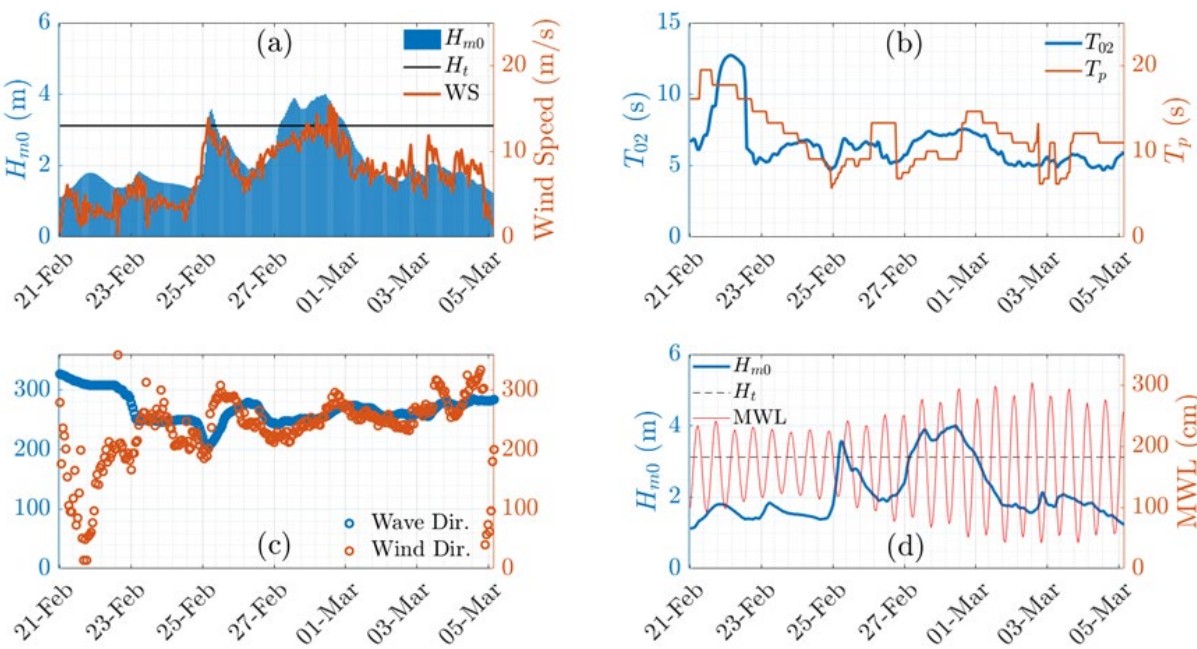

**Figure 10.** Significant wave height, wind speed (**a**), mean and peak periods (**b**), wind and wave directions (**c**), $H_{m0}$, and mean water level (**d**) evolution at point IP2 during a severe storm affecting southern strips of Tenerife island.

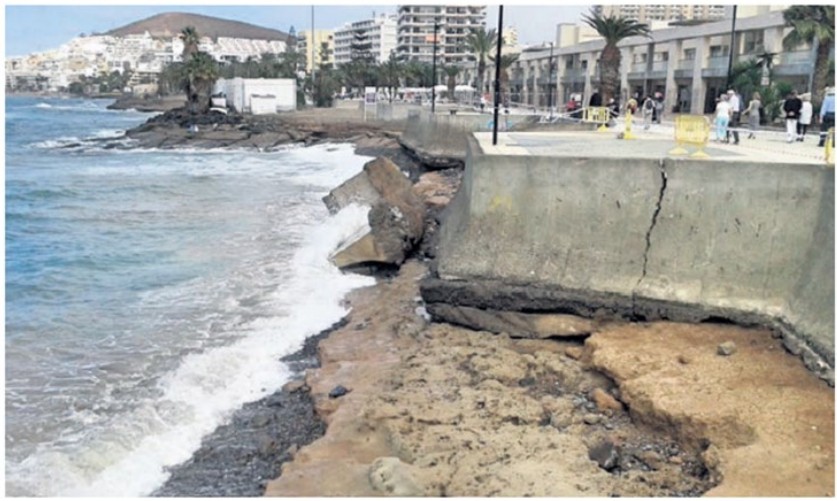

**Figure 11.** Photograph illustrating the effects of the storm Emma on Los Tarajales beach and promenade, municipality of Arona, Tenerife (Diario de Avisos, 4 March 2018).

## 5. Conclusions

Results of the long-term wind and wave datasets' analysis reveal a complex spatial pattern of wave storminess around the islands and in the southern flank of Tenerife, due to the intricacy of the coastline geometry, the presence of deep channels between islands, the high altitude and complex topography of the islands, and the sheltering effects exerted by each island over the others, depending on the directionality of the incident wave fields.

The most energetic events are detected on the western and northern flanks of the archipelago, while wave storms detected south or east of the archipelago are comparatively much milder, mainly due to sheltering effects against the direct action of storms approaching from any directional sector, except from those travelling from the SW and S sectors. In this sense, the points in the middle of the channels formed between the islands are special cases because they can receive waves from the NNW-NNE sector.

South of Tenerife, the severity of wave storms shows substantial spatial variability, with larger values in the southwestern and southern strips, exposed to relatively severe wave storms approaching from the SW, as well as to storms travelling through the G-TF channel from the NW.

The timing of extreme wave storms throughout the year in this area exhibits a marked seasonal character, occurring mainly during winter, between December and March, and becoming practically null in summer.

The directionality of wave storms at the southern flank of the island shows considerable spatial variability but reduced directional dispersion, mainly during stormy conditions, the period during which the channels on both sides of the island play a major role, with waves travelling through both channels but mainly from the NNW, through the G-T channel.

With regard to the role of sea water level during stormy conditions, it has been observed that the tidal range during the storms identified as damaging events was, in all cases, higher than the average value, highlighting the significant contribution of the tide to the impact of wave storms on the coastline.

In brief, the results evidence the importance of sheltering effects and the role of the G-T and GC-T channels, allowing NW and NE wave fields to reach the eastern and western edges of Southern Tenerife, with special relevance of the G-T channel through which waves generated by N-NW storms can reach tourist beaches located in the southwest coastal strip. Moderate storms from the W-SW-S sector predominate in the south-central area, while relatively weak storms propagating through the GC-T channel and associated with N-NE wind conditions dominate in the eastern coastal stretch.

Finally, the detailed analysis of specific severe storms highlights the vulnerability of tourist beaches on the southern and southwestern strips of Tenerife to unusual wave storms approaching from the S-SW sector, or from the NNW through the G-T channel, and the consequent strong socioeconomic impact of such events on this strategic beach tourism destination.

**Author Contributions:** Conceptualization, methodology and supervision, G.R.; software and validation, formal analysis, resources, and data curation, D.G.-M.; writing—original draft preparation, G.R.; writing—review and editing, G.R. All authors have read and agreed to the published version of the manuscript.

**Funding:** This research received no external funding.

**Institutional Review Board Statement:** Not applicable.

**Informed Consent Statement:** Not applicable.

**Data Availability Statement:** Datasets used in the study can be obtained from Puertos del Estado (http://www.puertos.es/es-es/oceanografia/Paginas/portus.aspx (accessed on 27 May 2021)).

**Acknowledgments:** The authors would like to thank Puertos del Estado for providing the meteoceanic data used in this work.

**Conflicts of Interest:** The authors declare no conflict of interest.

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
