# Peer review of "Spatiotemporal Variability of Extreme Wave Storms in a Beach Tourism Destination Area"

_geosciences, doi:10.3390/geosciences11060237_

Round 1
Reviewer 1 Report
The authors clearly explores the spatiotemporal variability of extreme wave storms around the Canary Archipelago by comparing wind and wave statistics at different locations. The results are very interesting to coastal engineers and marine scientists.
Author Response
The authors would like to acknowledge the time and effort made by the reviewer to suggest options for improvement in the manuscript.
Reviewer 2 Report
the Manuscript is very well structured and written; the methods are appropriate and well
described and the results are solid and clearly presented.
Below there are some minor comments that could improve the overall quality of the Manuscript.
About the wave storm concept and definition (Section 3.1), you can find a detailed description of a
procedure for the identification of wave storms in the following article
Salvadori, G., Tomasicchio, G.R., D’Alessandro, F. et al. Multivariate sea storm hindcasting and design:
the isotropic buoy-ungauged generator procedure. Sci Rep 10, 20517 (2020).
https://doi.org/10.1038/s41598-020-77329-y
Referencing this article could be useful to add scientific soundness to the procedure you adopted, since
the two procedures are fairly similar.
Moreover, to add scientific soundness to the usage of quantiles as a mean to reject storms of lower
intensities, the following Reference can be added:
Davies, G., Callaghan, D.P., Gravois, U., Jiang, W., Hanslow, D., Nichol, S., Baldock, T., 2017. Improved
treatment of non- stationary conditions and uncertainties in probabilistic models of storm wave climate.
Coast. Eng. 127, 1—19. https://doi.org/ 10.1016/j.coastaleng.2017.06.005
Finally, a Paragraph containing the Conclusions should be added at the end of the Manuscript.
Conclusions are useful because they present a summary of the major findings of the Manuscript and
because they draw future outlook or future directions for your work, therefore they add value to the work
described in the Manuscript.
Author Response
The authors wish to acknowledge the reviewer's time and effort in identifying weaknesses and suggesting options for improvement in the manuscript.

Reviewer 3 Report
Please find the attached file

Author Response
The authors wish to acknowledge the reviewer's time and effort in identifying weaknesses and suggesting options for improvement in the manuscript. (Please see the attachment).
